# Hypertension-Related Status and Influencing Factors among Chinese Children and Adolescents Aged 6~17 Years: Data from China Nutrition and Health Surveillance (2015–2017)

**DOI:** 10.3390/nu16162685

**Published:** 2024-08-13

**Authors:** Yuxiang Yang, Yuge Li, Hongtao Yuan, Zengxu Tang, Mulei Chen, Shuya Cai, Wei Piao, Jing Nan, Fusheng Li, Dongmei Yu, Xiang Gao

**Affiliations:** 1National Institute for Nutrition and Health, Chinese Center for Disease Control and Prevention, Beijing 100050, China; yxyang_ninhccdc@126.com (Y.Y.); liyuge1122@163.com (Y.L.); yuanht@ninh.chinacdc.cn (H.Y.); tangzx@ninh.chinacdc.cn (Z.T.); caisy@ninh.chinacdc.cn (S.C.); piaowei@ninh.chinacdc.cn (W.P.); nj13939012762@163.com (J.N.); 18246668833@163.com (F.L.); 2Chinese Center for Disease Control and Prevention, Beijing 100050, China; chenml@chinacdc.cn; 3Key Laboratory of Public Nutrition and Health, National Health Commission of the People’s Republic of China, Beijing 100050, China; 4Department of Nutrition and Food Hygiene, School of Public Health, Institute of Nutrition, Fudan University, Shanghai 200032, China

**Keywords:** children, adolescent, blood pressure, hypertension, influencing factor

## Abstract

Hypertension is currently highly prevalent worldwide and serves as one of the significant risk factors for chronic diseases and mortality. Adult hypertension can be traced back to, as well as prevented starting in, childhood and adolescence. However, due to the lack of surveillance among children and adolescents, the prevalence and influencing factors of hypertension-related conditions have not been well described. Hence, a total of 67,947 children and adolescents aged 6 to 17 from China Nutrition and Health Surveillance (2015–2017) were enrolled to describe the weighted average blood pressure level and the weighted prevalence of hypertension, pre-hypertension, and their distribution and to analyze the risk factors for hypertension and pre-hypertension among Chinese children and adolescents at a nationwide level. In summary, the weighted mean values of systolic blood pressure and diastolic blood pressure were 111.8 (95% CI, 111.2–112.5) mmHg and 66.5 (95% CI, 66.0–67.0) mmHg, respectively. The weighted prevalence of hypertension and pre-hypertension was 24.9% and 17.1%, respectively. Moreover, general obesity, overweight, and central obesity served as risk factors for hypertension and pre-hypertension among Chinese children and adolescents. The current study indicated that the prevalence of hypertension and pre-hypertension in Chinese children and adolescents was at a high level. Moreover, blood pressure screening should be further intensified for children and adolescents at a high risk of being overweight or obese.

## 1. Introduction

Hypertension (HTN) is a significant risk factor for cardiovascular and cerebrovascular diseases. The high rates of disability and death from hypertensive complications such as stroke, coronary heart disease, and heart failure pose a significant public health challenge worldwide [1,2]. According to the Global Report on Hypertension published by the World Health Organization in 2023, HTN affects one-third of adults all over the world, and the number of hypertensive patients has risen from 650 million in 1990 to 1.3 billion in 2019 [3]. In China, the issue of HTN is also urgent. According to results from the China Hypertension Survey (2012–2015), the prevalence of HTN among Chinese adults was 27.9% [4]. Compared with the past five rounds of China Hypertension Surveys, the overall prevalence of HTN among Chinese adults has shown an increasing trend [4,5].

HTN in adulthood can originate in childhood [6]. Studies have shown that individuals with persistently elevated blood pressure from childhood to adolescence are more likely to develop self-reported HTN in adulthood compared to those with normal blood pressure [7]. In 2010, the prevalence of elevated blood pressure among children and adolescents aged 7 to 17 in China was 16.1% for boys and 12.9% for girls [8]. Over the past two decades, the prevalence of HTN in children and adolescents has been increasing, and currently, millions of children worldwide have elevated blood pressure [9]. Another study also reported that around 4% of Chinese urban children and adolescents aged 6~17 years were confirmed to have hypertension [10]. However, the onset of HTN in childhood and adolescence is difficult to detect because the symptoms of elevated blood pressure have not yet manifested during this period [11]. Previous studies have shown that HTN and pre-hypertension (pre-HTN) were frequently undiagnosed in the pediatric population [12,13]. Therefore, understanding the prevalence of HTN-related status among Chinese children and adolescents is essential for developing new strategies for future HTN prevention, especially at a nationwide level [14]. Furthermore, the factors influencing HTN among Chinese children and adolescents are not fully understood, which is key to HTN identification and management [15]. Thus, identifying related influencing factors is equally urgent currently.

Based on China Nutrition and Health Surveillance (2015–2017), our present research aims to describe the weighted average blood pressure level and the weighted prevalence of HTN, pre-HTN, and their distribution among Chinese children and adolescents aged 6 to 17 years, and to analyze the risk factors for HTN and pre-HTN.

## 2. Materials and Methods

### 2.1. Participants

Participants were obtained from the China Nutrition and Health Surveillance (CNHS) 2015–2017, in which the CNHS of Children and Lactating Mothers was conducted from 2016 to 2017 (approved by the Ethics Committee of the Chinese Center for Disease Control and Prevention, Protocol Code: No. 201614). A previous study has detailed information concerning the study design, sampling method, and quality control process [16]. Briefly, a multistage stratified random sampling method was adopted in all 31 provinces of mainland China to obtain a representative sample of Chinese children and adolescents aged 6 to 17 years. The sample size of the respondents aged 6~17 years was based on the prevalence of overweight (4.5%) in 2013 and the non-response rate (10%). Finally, 71,035 respondents were sampled. The first stage of sampling was to select 275 survey sites randomly based on the considerations of regions and urbanization. Then, using a simple random sampling method, 2 subdistricts or 2 townships were selected for each survey site. Next, one elementary school and one middle school were randomly selected within each subdistrict or township, and one high school was randomly selected within each survey site directly. In the final stage of sampling, 1 class was randomly selected within each of the appropriate grade levels in the sampled schools, and 28 students were randomly selected from the sampled classes to complete the survey. At least 280 children and adolescents aged 6 to 17 were randomly selected from each survey site, 50/50 males and females.

After exclusion because of missing information on essential indicators, a total of 67,947 participants aged 6 to 17 years of age were included in the current study. All participants had signed informed consent before the survey by either themselves or their parents.

### 2.2. Information Collection

Using a standardized questionnaire, well-trained staff from the local Center for Disease Control and Prevention (CDC) collected information on demographic characteristics, socioeconomic factors, and lifestyle factors. Given the consideration of reading comprehension skills, the information of respondents in primary schools was collected by interviewing their parents/caregivers, whereas the information of respondents in junior high schools was collected by interviewing themselves.

Children, adolescents, and/or their parents/caregivers were also asked to complete a validated 57-item Food Frequency Questionnaire (FFQ) about dietary habits. The Dietary Approaches to Stop Hypertension (DASH) scores were calculated to assess the dietary status of each participant. Details concerning the calculation of dietary scores were available in our published study [17].

### 2.3. Anthropometric and Blood Pressure Measurement

All participants underwent anthropometric measurements according to standardized protocols by well-trained staff in the morning on an empty stomach during the investigation period. The height of the children and adolescents was measured using a TZG stadiometer in centimeters (cm) to the nearest 0.1 cm. Weight without heavy clothing and footwear was measured with a weighing scale (G&G TC–200 k, Changshu, China) in kilograms (kg) to the nearest 0.1 kg. Body mass index (BMI) was calculated as body weight (kg) divided by height squared (m^2^). The waist circumference (WC) was measured at the midpoint between the bottom of the rib cage and the top of the iliac crest twice using a soft tape, and the average value was used as the WC of participants. The waist-to-height ratio (WHtR) was calculated as the WC in centimeters divided by the height in centimeters.

BP measurements were performed using the automatic electronic oscillometric sphygmomanometer (OMRON HBP-1300, Tokyo, Japan) with an accuracy of 1 mmHg. Moreover, each sphygmomanometer was equipped with three types of measuring cuffs, namely SS size (12~18 cm), S size (17~22 cm), and M (22~32 cm). Investigators were asked to select the appropriate size of measuring cuff based on the respondent’s age and upper arm circumference. First, children and adolescents were seated at a table and were given 5 min to rest. After the resting period, BP was measured three times in a sitting position on the left arm at a heart level, with a 1 min interval. Eventually, the systolic blood pressure (SBP) and diastolic blood pressure (DBP) were represented by three-time average measurements of SBP and DBP.

### 2.4. Outcomes Definitions

The 2017 updated blood pressure references for Chinese children and adolescents aged 3~17 years were used to define normal BP, pre-HTN, and HTN [18]. Age-, sex-, and height-specific SBP and DBP percentile tables were used to define BP categories. Specifically, they were defined by year and by age, sex (males or females), and different height groups. Normal BP was defined as SBP and DBP of less than the 90th percentile; pre-HTN was defined as SBP and/or DBP from the 90th to less than the 95th percentile (or SBP/DBP ≥ 120/80 mmHg); and HTN was defined as SBP and/or DBP of at least the 95th percentile for sex, age, and height, respectively.

For general obesity, the participants were classed as normal, overweight, and obesity based on age- and sex-specific BMI percentile tables from the recommendations of the Working Group on Obesity in China [19]. Central obesity (abdominal obesity) was defined according to whose WHtR > 0.46 [20]. Other basic statuses included gender (males/females), age group (6~11/12~17), living area (urban/rural), geographical region (east/central/west), maternal education level (primary school or below/junior high school/senior high school or above) [21], and household income per capita (CNY < 10,000/CNY 10,000/CNY > 25,000/not given). Lifestyle factors included physical activity level (a daily average duration of moderate-to-vigorous physical activity achieved over 60 min, adequate/inadequate) [22], video time (a daily average duration of spending on TV, smartphone, or other types of electronic screens within a day, ≤2 h />2 h) [23], sleep duration (<7 h, 7~ h, or ≥9 h), and second-hand smoking exposure (yes/no). A family history of HTN was deemed as any immediate relatives (i.e., grandparents, parents, and siblings) who had HTN (yes/no). DASH scores were divided into quartiles (Q1–Q4).

### 2.5. Statistical Analysis

All values were weighted to address the multilevel stratified sampling design to represent the total population of Chinese children and adolescents aged 6 to 17 years based on the Chinese Census 2010. The survey weight used in this study was calculated based on the data published by the China National Bureau of Statistics in 2010, including post-stratification weights and sampling weights, to obtain national representativeness and the same population structures between survey samples and the national population. Categorical variables were reported as numbers and percentages. Weighted blood pressure level was reported as the mean with a 95% confidence interval using the PROC SURVEYMEANS in SAS software, and the statistical difference across groups was tested by two-tailed Student-*t* tests using the PROC SURVEYREG in SAS software. The prevalence of HTN and pre-HTN was reported as a percentage with a 95% confidence interval, and the statistical difference across groups was tested by Rao-Scott chi-squared tests using the PROC SURVEYFREQ in SAS software. Logistic regression was used for risk factors analysis using the PROC SURVEYLOGISTIC in SAS software. Results regarding pre-HTN were based on the exclusion of participants who were diagnosed by HTN in the current study.

Statistical analyses were conducted with SAS version 9.4 (SAS Institute, Inc., Cary, NC, USA). All tests were 2-sided, and *p* < 0.05 was considered statistically significant.

## 3. Results

### 3.1. Characteristics of Study Participants

In total, 67,947 children and adolescents aged 6 to 17 were included in the current analysis (Table 1). Among these participants, there were 7498 (11.0%) and 6186 (8.0%) children and adolescents identified as overweight and obese, respectively, and 16,020 (21.9%) as having central obesity. There were significant differences between males and females for all characteristics except geographical region and maternal education level. Males had a higher body mass index and waistline, had a lower age, and were more likely to have higher household income levels, a longer video time, and second-hand smoking exposure, to be less physically active, and to have a lower sleep duration, a family history of HTN, and lower DASH scores in comparison with females (*p* < 0.05).

### 3.2. Weighted Blood Pressure Level

Among Chinese children and adolescents 6~17 years of age, the weighted mean values of SBP and DBP were 111.8 (95% CI, 111.2–112.5) mmHg and 66.5 (95% CI, 66.0–67.0) mmHg, respectively (Table 2). The weighted mean values of SBP and DBP for males increased with age. For females, the weighted mean values of SBP peaked at puberty and then plateaued (Figure 1, Appendix A).

### 3.3. Weighted Prevalence of HTN and Pre-HTN

The weighted prevalence of HTN among Chinese children and adolescents 6~17 years of age was 24.9%, and the weighted prevalence of pre-HTN was 17.1% (Table 2). The overall crude prevalences of HTN and pre-HTN were 24.3% and 15.8%, respectively (Appendix A).

The weighted prevalence of HTN was over two times higher in general obese individuals compared with those of normal weight (50.4% versus 21.3%, *p* < 0.0001) and was higher in children with central obesity than in those without (36.4% versus 21.7%, *p* < 0.0001). The weighted prevalence of HTN was also higher among children and adolescents with medium and lower household incomes and longer sleep duration. The weighted prevalence of pre-HTN was higher in males than in females (19.7% versus 14.1%, *p* < 0.0001) and was higher among those with obesity and overweight individuals compared with those of normal weight (obesity: 17.5% versus 16.3%, overweight: 22.1% versus 16.3%, *p* < 0.0001). Moreover, the weighted prevalence of pre-HTN was also shown to be higher among the central region, longer video time, and shorter sleep duration (Table 2). The weighted prevalence of HTN decreased with the increase in age for both males and females. The weighted prevalence of pre-HTN for males increased rapidly at puberty (Figure 2).

Notably, the prevalence of HTN and pre-HTN varied among provinces (Table 3). Hebei province, Chongqing city, and Jiangsu province ranked in the top three for an HTN prevalence rate of 38.7%, 37.2%, and 33.2%, respectively. Anhui province, Liaoning province, and Jilin province ranked in the top three for a pre-HTN prevalence rate of 23.5%, 20.5%, and 20.0%, respectively.

### 3.4. Multivariable Risk Assessment

In the weighted multivariate logistic regression, compared with reference groups, females (OR = 1.09), overweight (OR = 1.73), obesity (OR = 3.30), central obesity (OR = 1.23), living in a rural area (OR = 1.28), and longer sleep duration (≥9 h, OR = 1.26) were all positively associated with HTN, while the senior age group (OR = 0.79) and higher household income (CNY >25,000) (OR = 0.79) were negatively associated with HTN (all *p* value < 0.05). Meanwhile, the senior age group (OR = 1.42), overweight (OR = 1.78), and obesity (OR = 2.07) were significantly associated with pre-HTN, while females (OR = 0.66) were negatively associated with pre-HTN (all *p* value < 0.05) (Figure 3). Further information is available in Appendix A.

## 4. Discussion

In this nationally representative cross-sectional study, the status of HTN and pre-HTN among Chinese children and adolescents 6~17 years were estimated. It showed that 24.9% of Chinese children and adolescents had HTN, whereas 17.1% had pre-HTN. The prevalence of HTN and pre-HTN were significantly higher among those who were obese or overweight. The central region was shown to have the highest prevalence of HTN and pre-HTN compared with the prevalence in eastern and western regions. However, when compared by provinces, Hebei province, which was in the eastern region, had the highest HTN prevalence, whereas Anhui province, which was in the central region, had the highest pre-HTN prevalence. General obesity, overweight, and central obesity were shown to be risk factors for HTN and pre-HTN.

Compared with the results of CNHS 2010–2012, which also used the multistage stratified random sampling method, the SBP level among Chinese children and adolescents has increased from 101 mmHg to 111.8 mmHg, and the DBP level has increased from 65 mmHg to 66.5 mmHg [24]. Previous studies have shown less consistency between the results obtained from different blood pressure measuring devices among children and adolescents, with electronic sphygmomanometers having significantly higher results than mercury column sphygmomanometers [25]. Since electronic sphygmomanometers were used in our study and mercury sphygmomanometers were used in CNHS 2010–2012, this may have led to the difference in results. Therefore, researchers should consider the differences in measurement protocols and measurement methods, and when using different devices, the results should be corrected to obtain comparable results and scientific conclusions [26]. In addition, because the prevalence of obesity or overweight has increased among Chinese children in the past decade [27], this could lead to dramatic changes in blood pressure levels.

In this study, we found that SBP peaked in girls during puberty and then leveled off, whereas, in boys, systolic blood pressure continued to increase throughout the child–adolescent stage and was higher than that of girls after puberty, which is consistent with the results of Chinese and international studies [28,29]. The underlying physiological mechanism is that the hypothalamus–pituitary–gonad axis (HPGA) secretes hormones, such as growth hormone, to promote growth in children and adolescents during the pubertal stage, leading to increased blood pressure levels. Estrogen secretion in girls reaches the level of adult females at this stage, leading to the difference in blood pressure and the prevalence of HTN and pre-HTN between boys and girls [30,31,32]. Since puberty is a critical period of growth and development for children and adolescents, more attention should be paid to the relationship between different pubertal onset phases and blood pressure to obtain more relevant conclusions [33].

Our results showed that the prevalence of HTN in children and adolescents is generally high in the eastern regions and low in the western regions. The rapid increase in economic growth in recent years in eastern regions has resulted in the overconsumption of high-energy-density food, and this change in dietary patterns can lead to overweight and obesity, which further contribute to the development of HTN [34]. In addition, children and adolescents in developed areas suffer from more significant academic stress, together with an unhealthy lifestyle, which most likely contributes to the development of HTN [35,36]. These results suggest that increased attention should be paid to factors such as the psychological status of children and adolescents on health in HTN management with different strategies in HTN management.

As overweight and obesity are spiking among Chinese children and adolescents aged 6~17 years, the prevalence has reached 11.1% and 7.9% for them, respectively [37]. This condition has posed a significant threat to HTN-related status among Chinese children and adolescents [9,38,39]. The underlying physiological mechanism is that overweight and obesity lead to an increased secretion of adipocytokine, sympathetic nervous system excitation, activation of the renin–angiotensin–aldosterone system, lipid or insulin abnormalities, and alterations in the leptin pathway, which further leads to decreased vascular compliance, increased cardiac output, or insulin resistance [40]. A meta-analysis in Africa showed that the prevalence of HTN was closely related to body mass index (BMI), and the prevalence of high blood pressure in obese children (30.8%) was nearly six times higher than that of normal-weight children (5.5%) [41]. DONG et al. found that the prevalence of high blood pressure in overweight and obese children and adolescents in China was also significantly higher than that of children and adolescents with normal body weight and that the elimination of overweight and obesity could reduce the number of children and adolescents suffering from high blood pressure by 14.4% [8]. In addition, the effect of obesity on blood pressure may be related to fat distribution, and central obesity is more likely to lead to HTN as well as the aggregation of cardiovascular disease risk factors other than general obesity [42]. The above conclusions were consistent with the findings of our study, which all indicated that the prevention of overweight and obesity in children and adolescents has become an essential approach for HTN control.

For other influencing factors, we also observed that living in rural areas and having longer sleep durations were positively associated with HTN, while higher household income was negatively associated with HTN. Similar to the previous study, the prevalence of HTN in Chinese rural areas has significantly spiked recently [43]. Rural residents in China had fewer opportunities to access high-quality health-care service and were more prone to lower household income, which might also impact their HTN-related health awareness and intervention [43]. Regarding sleep, a cross-sectional study among Chinese adults showed that over 9 h of sleep duration had a positive association with HTN, which might be interpreted by sleep fragmentation, immune, and other chronic disease status [44,45]. However, the above potential mechanisms among children and adolescents needed further clarification.

In this study, the prevalence of HTN and pre-HTN from Chinese children and adolescents was higher than that from other nations. Data from a Canadian national sample from 2007 to 2015 showed that the prevalence of pre-HTN and HTN (systolic/diastolic blood pressure ≥ P90) in children and adolescents aged 6 to 18 years was 5.8% [46]. A sample survey conducted in Italy in 2007 showed that the prevalence of HTN among children and adolescents aged 6 to 17 years was 6.2% [47]. Using data from the Korea National Health and Nutrition Examination Survey (KNHANES) on 7804 children and adolescents aged 10 to 18 years, the researchers found that in 2013–2015, pre-HTN and HTN prevalence were 8.8% and 9.0%, respectively [48]. In addition to race being a major influencing factor, differences in the diagnostic criteria utilized are also a significant factor in the disparity in results [49,50].

Unlike the diagnostic criteria for HTN in adults, the criteria for determining HTN in children and adolescents are based on the blood pressure percentile reference values of the general healthy population because there are no exact indicators of cardiovascular health or disease outcomes in adulthood that are associated with blood pressure status in childhood [51]. The references used in this study are all based on the 95th percentile of blood pressure reference values for children and adolescents, so the detection rate of HTN should be around 5% [52], but the results obtained were much higher than 5%. Therefore, the use of the 95th percentile as a criterion for determining HTN may lead to the overestimation of the prevalence of HTN. Further cohort studies should be conducted to determine the diagnostic threshold of blood pressure oriented to disease outcomes, which will be more clinically significant for diagnosing HTN among children and adolescents. In addition, it is recommended that standardized blood pressure data, measurement protocols, and diagnostic criteria with universal applicability be established worldwide and in different regions to make results more comparable across studies.

The current study has several limitations. First, the causality between the influencing factors and HTN cannot be confirmed due to the cross-sectional design. Second, our blood pressure data were based on three averaged measurements taken during a single visit instead of across more than three different occasions. Third, although this study included as many influencing factors as possible, there are still factors that were not taken into account, such as birth outcomes and gestational exposures, atmospheric pollution, noise exposure, and so on.

## 5. Conclusions

This study described the weighted average blood pressure level, the weighted prevalence of HTN and pre-HTN, and their distribution among Chinese children and adolescents aged 6 to 17 years, and found that general obesity, overweight, and central obesity were risk factors for HTN and pre-HTN among Chinese children and adolescents. Our study supported that blood pressure screening should be further intensified for children and adolescents at high risk, such as being overweight or obese or living in rural areas.

## Figures and Tables

**Figure 1 nutrients-16-02685-f001:**
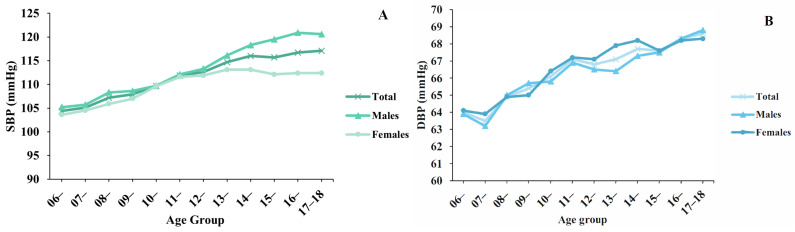
Weighted blood pressure level among Chinese children and adolescents 6~17 years of age by gender. (**A**) SBP and (**B**) DBP.

**Figure 2 nutrients-16-02685-f002:**
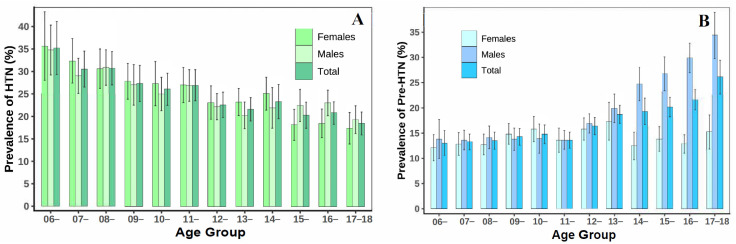
Weighted prevalence of HTN status among Chinese children and adolescents 6~17 years of age by gender. (**A**) HTN and (**B**) pre-HTN.

**Figure 3 nutrients-16-02685-f003:**
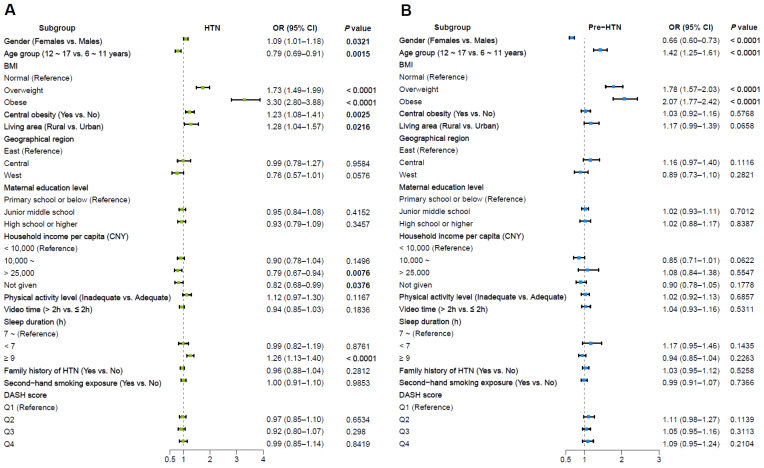
Multivariable adjusted odds ratios for HTN and pre-HTN among Chinese children and adolescents 6~17 years of age by gender. (**A**) HTN and (**B**) pre-HTN. (The *p* values in bold indicate *p* < 0.05).

**Table 1 nutrients-16-02685-t001:** Characteristics of study participants.

Characteristics, n (%)	Total	Gender		*p* Value
		Males	Females	
Overall	67,947 (100.0)	33,882 (49.9)	34,065 (50.1)	
Age group (years)				<0.0001
6~11	37,829 (48.5)	18,833 (49.1)	18,996 (47.9)	
12~17	30,118 (51.5)	15,049 (50.9)	15,069 (52.1)	
BMI				<0.0001
Normal weight	54,263 (80.9)	25,767 (77.1)	28,496 (85.3)	
Overweight	7498 (11.0)	4288 (12.8)	3210 (9.1)	
Obese	6186 (8.0)	3827 (10.1)	2359 (5.7)	
Central obesity				<0.0001
No	51,927 (78.1)	24,853 (75.1)	27,074 (81.5)	
Yes	16,020 (21.9)	9029 (24.9)	6991 (18.5)	
Living area				0.4189
Urban	32,594 (47.3)	16,247 (47.2)	16,347 (47.4)	
Rural	35,353 (52.7)	17,635 (52.8)	17,718 (52.6)	
Geographical region				0.5588
East	23,235 (39.9)	11,603 (39.9)	11,632 (40.0)	
Central	21,397 (27.6)	10,667 (27.7)	10,730 (27.5)	
West	23,315 (32.4)	11,612 (32.4)	11,703 (32.5)	
Maternal education level				0.746
Primary school or below	18,241 (27.4)	9172 (27.3)	9069 (27.6)	
Junior middle school	41,051 (60.6)	20,417 (60.6)	20,634 (60.5)	
High school or higher	8655 (12.0)	4293 (12.1)	4362 (11.9)	
Household income per capita (CNY)				<0.0001
<10,000	9859 (13.4)	4767 (13.1)	5092 (13.8)	
10,000~	10,729 (14.3)	5516 (14.7)	5213 (13.9)	
25,000~	4265 (5.8)	2274 (6.4)	1991 (5.1)	
Not given	43,094 (66.5)	21,325 (65.8)	21,769 (67.2)	
Physical activity level				<0.0001
Adequate	43,769 (66.2)	23,128 (70.1)	20,641 (61.8)	
Inadequate	24,178 (33.8)	10,754 (29.9)	13,424 (38.2)	
Video time (h)				<0.0001
≤2	20,847 (31.2)	9564 (27.1)	11,283 (30.8)	
>2	47,100 (73.6)	24,318 (72.9)	22,782 (69.2)	
Sleep duration (h)				0.0003
<7	2154 (4.9)	947 (4.3)	1207 (5.7)	
7~	21,316 (37.4)	10,480 (37.0)	10,836 (37.9)	
≥9	44,477 (57.7)	22,455 (58.7)	22,022 (56.5)	
Family history of HTN				<0.0001
No	45,410 (66.8)	23,016 (68.1)	22,394 (65.3)	
Yes	22,537 (33.2)	10,886 (31.9)	11,671 (34.7)	
Second-hand smoking exposure				0.0010
No	38,955 (56.8)	18,851 (55.4)	20,104 (58.4)	
Yes	28,992 (43.2)	15,031 (44.6)	13.961 (41.6)	
DASH score				<0.0001
Q1	18,131 (28.9)	9642 (31.0)	8489 (26.6)	
Q2	13,124 (19.4)	6628 (19.3)	6496 (19.5)	
Q3	18,900 (27.0)	9231 (26.5)	9669 (27.7)	
Q4	17,792 (24.6)	8381 (23.2)	9411 (26.2)	

All values were weighted to represent the total population of Chinese children and adolescents 6~17 years of age based on the Chinese Census 2010. Values of polytomous variables may not sum to 100% due to rounding. Continuous variables were described as means with standard errors. Categorical variables were described as amounts with percentages. Values of polytomous variables may not sum to 100% because of rounding. Abbreviation: BMI, body mass index; DASH: Dietary Approaches to Stop Hypertension.

**Table 2 nutrients-16-02685-t002:** Weighted blood pressure level and prevalence of pre-HTN and HTN by characteristics.

Characteristics	N	Blood Pressure (mmHg)		HTN Classification (%)		
		SBP	DBP	Normal	Pre-HTN	HTN
Overall	67,947	111.8 (111.2–112.5)	66.5 (66.0–67.0)	58.0 (55.5–60.5)	17.1 (16.1–18.0)	24.9 (22.7–27.2)
Gender						
Males	33,882	113.5 (112.7–114.2)	66.4 (65.8–66.9)	55.3 (52.6–57.9)	19.7 (18.3–21)	25.0 (22.8–27.3)
Females	34,065	110.0 (109.4–110.6)	66.7 (66.1–67.2)	61.1 (58.4–63.7)	14.1 (13.3–15)	24.8 (22.3–27.2)
*p* value		<0.0001	0.0026		<0.0001	0.1628
BMI						
Normal	54,263	110.4 (109.8–111.0)	66.1 (65.5–66.6)	62.4 (59.7–65.1)	16.3 (15.3–17.4)	21.3 (18.9–23.6)
Overweight	7498	116.2 (115.5–117.0)	67.8 (67.3–68.3)	44.7 (41.7–47.7)	22.1 (20.5–23.8)	33.2 (30.2–36.2)
Obese	6186	120.2 (119.1–121.3)	69.2 (68.5–69.9)	32.1 (29.3–34.9)	17.5 (15.9–19.1)	50.4 (47.1–53.8)
*p* value		<0.0001	<0.0001		<0.0001	<0.0001
Central obesity						
No	51,927	110.6 (110.0–111.2)	66.1 (65.6–66.7)	61.7 (58.9–64.4)	16.7 (15.5–17.8)	21.7 (19.2–24.2)
Yes	16,020	116.2 (115.3–117.1)	67.9 (67.4–68.4)	45.0 (42.6–47.4)	18.6 (17.5–19.6)	36.4 (34.0–38.8)
*p* value		<0.0001	<0.0001		<0.0001	<0.0001
Living area						
Urban	32,594	112.5 (111.5–113.5)	66.2 (65.5–66.9)	59.9 (56.7–63.1)	17.1 (15.6–18.5)	23.0 (20.4–25.6)
Rural	35,353	111.3 (110.5–112.1)	66.8 (66.1–67.5)	56.3 (52.7–59.9)	17.1 (15.8–18.3)	26.6 (23.2–30.1)
*p* value		0.0653	0.2612		0.0822	0.4759
Geographical region						
East	23,235	112.7 (111.5–113.9)	66.7 (65.7–67.6)	57.6 (53.1–62.1)	16.9 (15.1–18.7)	25.5 (21.9–29.1)
Central	21,397	112.7 (111.7–113.7)	67.1 (66.3–67.9)	54.4 (50.8–58.0)	18.6 (17.4–19.9)	27.0 (23.5–30.4)
West	23,315	110.1 (109.0–111.1)	65.8 (65.0–66.6)	61.6 (57.1–66.0)	16.0 (14.3–17.7)	22.5 (18.0–26.9)
*p* value		0.0004	0.0856		0.0342	0.2913
Maternal education level						
Primary school or below	18,241	112.0 (111.3–112.8)	66.9 (66.4–67.5)	58.5 (55.6–61.4)	17.1 (16.0–18.2)	24.4 (21.6–27.2)
Junior middle school	41,051	111.7 (111.0–112.4)	66.4 (65.9–67.0)	57.7 (55.0–60.5)	16.9 (15.8–18.0)	25.3 (22.9–27.8)
High school or higher	8655	112.0 (110.9–113.1)	66.1 (65.3–66.8)	58.3 (54.8–61.7)	17.8 (15.3–20.3)	24.0 (20.9–27.0)
*p* value		0.4504	0.0473		0.8529	0.5696
Household income per capita						
<10,000	9859	110.9 (109.9–111.9)	66.4 (65.7–67.2)	55.5 (51.8–59.2)	16.4 (14.6–18.3)	28.1 (24.5–31.6)
10,000~	10,729	111.8 (111.0–112.5)	66.2 (65.6–66.9)	57.4 (53.7–61.2)	15.9 (14.5–17.3)	26.7 (23.4–29.9)
>25,000	4265	112.9 (111.8–114.0)	66.2 (65.3–67.0)	55.9 (52.3–59.4)	20.3 (17.6–23.0)	23.9 (21.1–26.6)
Not given	43,094	111.9 (111.2–112.7)	66.6 (66.0–67.2)	58.8 (56.1–61.6)	17.2 (16.0–18.4)	24.0 (21.5–26.5)
*p* value		0.0165	0.5653		0.1276	0.013
Physical activity level						
Adequate	43,769	111.9 (111.2–112.6)	66.2 (65.7–66.8)	58.7 (56.2–61.3)	17.2 (15.9–18.4)	24.1 (22.0–26.2)
Inadequate	24,178	111.8 (111.0–112.6)	67.1 (66.5–67.7)	56.6 (53.2–59.9)	16.9 (15.9–17.9)	26.5 (23.2–29.9)
*p* value		0.7982	0.0015		0.7015	0.0807
Video time (h)						
≤2	20,847	110.9 (110.2–111.6)	66.4 (65.9–66.9)	57.8 (54.9–60.8)	15.8 (14.5–17.2)	26.3 (23.9–28.8)
>2	47,100	112.2 (111.5–112.9)	66.8 (66.0–67.1)	58.1 (55.5–60.7)	17.6 (16.5–18.6)	24.3 (22.0–26.7)
*p* value		<0.0001	0.2831		0.0195	0.0813
Sleep duration (h)						
<7	2154	116.7 (115.5–117.9)	68.4 (67.3–69.4)	56.5 (51.3–61.7)	22.8 (19.1–26.5)	20.6 (17.3–24.0)
7~	21,316	114.7 (114.0–115.4)	67.2 (66.7–67.8)	59.8 (57.2–62.3)	19.7 (18.2–21.1)	20.5 (18.6–22.5)
≥9	44,477	109.6 (108.8–110.3)	65.9 (65.3–66.5)	57.0 (54.0–60.0)	14.9 (13.9–15.9)	28.1 (25.2–31.0)
*p* value		<0.0001	<0.0001		<0.0001	<0.0001
Family history of HTN						
No	45,410	111.5 (110.8–112.2)	66.5 (66.0–67.0)	58.2 (55.4–60.9)	16.7 (15.7–17.7)	25.1 (22.5–27.7)
Yes	22,537	112.5 (111.8–113.3)	66.5 (66.0–67.1)	57.7 (55.1–60.3)	17.8 (16.5–19.0)	24.5 (22.5–26.6)
*p* value		0.0001	0.9085		0.1207	0.5758
Second–hand smoking exposure						
No	38,955	111.7 (111.0–112.3)	66.7 (66.1–67.2)	57.9 (55.2–60.5)	17.0 (15.8–18.1)	25.1 (22.7–27.6)
Yes	28,992	112.1 (111.4–112.8)	66.3 (65.8–66.8)	58.2 (55.5–60.9)	17.2 (16.2–18.2)	24.6 (22.3–26.9)
*p* value		0.0697	0.0524		0.8430	0.5695
DASH score						
Q1	18,131	111.6 (110.7–112.5)	66.8 (66.1–67.4)	58.1 (54.2–61.9)	16.4 (15.1–17.8)	25.5 (21.7–29.2)
Q2	13,124	111.7 (111.0–112.5)	66.4 (65.9–67.0)	57.5 (55.0–60.0)	17.7 (16.2–19.2)	24.8 (22.7–26.9)
Q3	18,900	111.8 (111.1–112.5)	66.5 (65.8–67.1)	58.9 (56.0–61.8)	17.1 (15.9–18.4)	23.9 (21.4–26.4)
Q4	17,792	112.2 (111.5–112.9)	66.3 (65.8–66.9)	57.4 (54.4–60.4)	17.3 (16.1–18.5)	25.4 (22.7–28.0)
*p* value		0.4316	0.4646		0.5330	0.6939

All values were weighted to represent the total population of Chinese children and adolescents 6~17 years of age based on the Chinese Census 2010. Values of polytomous variables may not sum to 100% due to rounding. Data are represented as mean or percentage (95% confidence interval). Abbreviation: BMI, body mass index; SBP, systolic blood pressure; DBP, diastolic blood pressure; DASH: Dietary Approaches to Stop Hypertension.

**Table 3 nutrients-16-02685-t003:** Weighted prevalence of HTN and pre–HTN by provinces.

Province	N	HTN			*p* Value	Pre–HTN			*p* Value
		Total	Males	Females		Total	Males	Females	
Beijing City	1380	25.6 (19.8–31.3)	27.3 (22.1–32.6)	23.7 (14.7–32.6)	0.4979	16.4 (14.9–17.9)	19.1 (15.9–22.3)	13.4 (11.3–15.6)	0.0309
Tianjin City	1363	31.6 (25.2–37.9)	35.9 (32.4–39.5)	26.9 (16.8–36.9)	0.0032	18.7 (17.2–20.2)	24.1 (21.7–26.5)	12.9 (11.9–13.9)	<0.0001
Hebei	3050	38.7 (25.4–51.9)	35.8 (22.8–48.8)	42.0 (26.4–57.5)	0.262	19.5 (17.3–21.6)	23.7 (20.0–27.5)	14.5 (12.3–16.8)	0.0234
Shanxi	2404	22.5 (13.2–31.8)	24.3 (12.8–35.9)	20.4 (12.9–28.0)	0.2446	15.3 (10.0–20.5)	18.2 (10.6–25.8)	11.9 (8.4–15.4)	0.0018
Inner Mongolia Autonomous Region	2142	22.8 (16.4–29.2)	21.5 (15.9–27.2)	24.3 (15.6–33.0)	0.4232	15.5 (12.9–18.2)	18.6 (15.3–21.8)	12.0 (9.1–15.0)	<0.0001
Liaoning	1059	20.5 (16.9–24.1)	19.7 (15.5–23.9)	21.4 (18.2–24.7)	0.0021	20.5 (18.3–22.7)	26.4 (22.6–30.1)	14.1 (12.9–15.4)	<0.0001
Jilin	1321	23.8 (17.4–30.2)	27.3 (21.3–33.2)	20.2 (13.5–27.0)	<0.0001	20.0 (16.3–23.7)	25.4 (16.7–34.2)	14.4 (13.1–15.6)	0.0197
Heilongjiang	2871	31.6 (21.9–41.2)	31.3 (23.3–39.4)	31.9 (20.3–43.4)	0.799	18.5 (16.7–20.3)	23.6 (21.7–25.6)	12.8 (9.8–15.7)	<0.0001
Shanghai City	1006	26.1 (22.0–30.2)	28.1 (21.4–34.7)	23.9 (19.9–27.9)	0.3293	18.5 (14.1–22.9)	21.6 (17.2–26.0)	15.0 (10.2–19.9)	0.0009
Jiangsu	2985	33.2 (30.1–36.3)	32.6 (28.8–36.3)	34.0 (31.0–37.0)	0.2477	18.5 (15.9–21.2)	23.0 (19.9–26.0)	13.4 (11.0–15.8)	<0.0001
Zhejiang	2620	26.0 (21.7–30.3)	26.4 (22.2–30.6)	25.4 (20.6–30.3)	0.5282	15.7 (13.0–18.4)	17.4 (12.9–21.9)	13.7 (12.9–14.4)	0.0039
Anhui	2875	26.7 (24.0–29.4)	26.3 (21.9–30.7)	27.1 (24.1–30.1)	0.7971	23.5 (20.2–26.7)	29.1 (23.1–35.1)	17.2 (15.6–18.9)	<0.0001
Fujian	1824	22.7 (17.8–27.6)	23.8 (17.9–29.7)	21.4 (15.7–27.1)	0.5099	17.1 (15.0–19.1)	21.5 (18.5–24.4)	12.1 (9.6–14.5)	<0.0001
Jiangxi	2655	26.8 (18.9–34.6)	27.6 (20.4–34.9)	25.8 (17.1–34.4)	0.0748	19.4 (17.7–21.2)	20.7 (17.9–23.4)	18.0 (15.1–21.0)	0.2035
Shandong	3465	29.2 (17.0–41.5)	27.6 (14.2–40.9)	31.1 (19.8–42.4)	0.1765	18.8 (17.2–20.5)	21.0 (17.6–24.4)	16.3 (15.1–17.6)	0.1461
Henan	3664	31.5 (24.0–38.9)	30.0 (22.9–37.1)	33.1 (25.0–41.2)	0.0388	18.7 (17.1–20.3)	23.0 (20.5–25.6)	13.7 (12.1–15.3)	<0.0001
Hubei	2573	28.0 (23.9–32.2)	29.3 (26.4–32.2)	26.6 (19.8–33.5)	0.4322	16.7 (14.4–19.0)	17.4 (13.7–21.1)	15.9 (13.7–18.1)	0.2569
Hunan	3034	19.6 (11.5–27.7)	19.6 (13.0–26.2)	19.6 (9.4–29.7)	0.9892	16.8 (13.5–20.1)	15.6 (11.8–19.5)	18.1 (11.4–24.8)	0.5727
Guangdong	3445	14.9 (13.0–16.9)	17.4 (14.4–20.5)	12.1 (9.3–14.9)	0.0185	14.1 (8.6–19.6)	15.8 (8.6–23.0)	12.3 (8.0–16.7)	0.1046
Guangxi Zhuang Autonomous Region	2385	26.4 (17.5–35.3)	23.5 (16.5–30.6)	29.7 (18.6–40.8)	<0.0001	19.4 (16.5–22.3)	21.3 (17.9–24.7)	17.2 (13.5–20.9)	0.2328
Hainan	1038	22.5 (13.7–31.3)	25.5 (17.9–33.0)	19.2 (8.5–29.9)	0.1562	13.6 (11.7–15.4)	15.0 (13.5–16.6)	11.9 (7.7–16.1)	0.1819
Chongqing City	1628	37.2 (18.6–55.8)	38.0 (19.7–56.2)	36.4 (17.1–55.8)	0.6183	15.4 (13.1–17.8)	15.1 (12.3–17.8)	15.8 (13.7–18.0)	0.7818
Sichuan	3593	18.6 (15.5–21.7)	17.3 (12.3–22.4)	19.9 (16.2–23.7)	0.4588	18.1 (12.7–23.5)	22.2 (14.3–30.2)	13.5 (9.9–17.2)	0.0069
Guizhou	2410	24.8 (13.9–35.8)	25.4 (15.8–35.1)	24.1 (11.4–36.9)	0.6128	17.2 (14.3–20.0)	18.6 (15.7–21.5)	15.5 (12.0–19.0)	0.0269
Yunnan	3336	17.2 (11.6–22.9)	16.6 (11.7–21.5)	18.0 (11.1–24.8)	0.4336	13.7 (11.2–16.2)	15.2 (11.6–18.7)	12.1 (9.2–15.0)	0.0973
Shaanxi	1836	14.4 (11.2–17.6)	12.9 (9.8–16.0)	16.2 (11.8–20.6)	0.0881	13.5 (12.0–15.1)	17.0 (13.0–20.9)	9.6 (7.9–11.3)	0.0055
Gansu	1594	24.6 (15.6–33.7)	26.5 (15.7–37.4)	22.4 (14.7–30.1)	0.1816	16.8 (15–18.5)	18.6 (16.0–21.3)	14.6 (11.3–18.0)	0.1822
Qinghai	1567	14.9 (7.3–22.5)	14.4 (7.3–21.5)	15.4 (7.0–23.8)	0.4674	14.0 (9.7–18.4)	16.0 (10.3–21.7)	11.7 (8.2–15.2)	0.042
Ningxia Hui Autonomous Region	1508	15.7 (9.4–21.9)	14.8 (7.8–21.9)	16.7 (9.5–23.8)	0.6049	13.0 (8.7–17.4)	14.0 (9.6–18.4)	12.0 (7.6–16.3)	0.1333
Xinjiang Uyghur Autonomous Region	1265	13.5 (6.2–20.8)	13.5 (6.2–20.9)	13.4 (6.1–20.7)	0.8144	9.1 (6.2–11.9)	11.3 (7.7–14.8)	6.7 (4.3–9.1)	0.0002

Data are represented as percentages (95% CI). Due to the insufficient number of participants from the Tibet autonomous region, it was not presented in this table.

## Data Availability

According to the policy of the National Institute for Nutrition and Health, Chinese Center for Disease Control and Prevention, data related to this research are not allowed to be disclosed.

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
