# Peer review of "Hypertension-Related Status and Influencing Factors among Chinese Children and Adolescents Aged 6~17 Years: Data from China Nutrition and Health Surveillance (2015–2017)"

_nutrients, 2024, doi:10.3390/nu16162685_

Round 1
Reviewer 1 Report
Comments and Suggestions for Authors
Dear Authors,
Thank you for giving me the opportunity to review your manuscript which, in an extended paper, evaluates blood pressure levels and HTN prevalence in Chinese children and adolescents.
Here are my comments on your paper:
Chapter INTRODUCTION: HTN is well described, as well as how previous studies find various related outcomes. This chapter presents a good structure and organization of the mentioned data. The hypothesis and objectives are well presented.
Chapter MATERIALS AND METHODS: Good structure, well described.
RESULTS Chapter: Demographic information and results of questionnaires and measurements administered to children and adolescents are well systematized and explained. Figure 3 is not well visualized.
DISCUSSION CHAPTER: Results are well explained and include a good comparison with other similar studies.
Explain more clearly why HTN prevalence decreased with increasing age and pre-HTN increased with increasing age.
The subchapter "Limitations" presents some weaknesses of the present study in a very candid way.
CONCLUSIONS chapter: the aim of this study is achieved and the results are well presented.
Good luck!
Author Response
Comments 1: Thank you for giving me the opportunity to review your manuscript which, in an extended paper, evaluates blood pressure levels and HTN prevalence in Chinese children and adolescents.
Response 1: Dear reviewer, sincerely thank you for sparing your precious time on our manuscript entitled "Hypertension-Related Status and Influencing Factors Among Chinese Children and Adolescents Aged 6~17 Years: Data from China Nutrition and Health Surveillance (2015–2017)." All the suggestions and comments are valuable and helpful in improving our work and further research. Based on your professional suggestions, we have made extensive changes to our manuscript.
Comments 2: Chapter INTRODUCTION: HTN is well described, as well as how previous studies find various related outcomes. This chapter presents a good structure and organization of the mentioned data. The hypothesis and objectives are well presented.
Response 2: Dear reviewer, thank you for your positive comments!
Comments 3: Chapter MATERIALS AND METHODS: Good structure, well described.
Response 3: Dear reviewer, thank you for your positive comments!
Comments 4: RESULTS Chapter: Demographic information and results of questionnaires and measurements administered to children and adolescents are well systematized and explained. Figure 3 is not well visualized.
Response 4: Dear reviewer, thank you for your suggestions. In this version, we have already updated Figure 3 (penal A and penal B) in the manuscript; please see it in the new version.
Comments 5: DISCUSSION CHAPTER: Results are well explained and include a good comparison with other similar studies.
Response 5: Dear reviewer, thank you for your positive comments!
Comments 6: Explain more clearly why HTN prevalence decreased with increasing age and pre-HTN increased with increasing age.
Response 6: Dear reviewer, thank you for the detailed review and valuable comments on our paper. We have noted the issue of the variation in prevalence with age that you mentioned, and this phenomenon poses a challenge for interpreting our results. We have conducted a preliminary analysis of this phenomenon but have not yet found a definitive explanation. One possible reason is that the reference we used may not fully apply to our sample population, but this still requires further verification. We will continue to focus on this topic in further study.
Comments 7: The subchapter "Limitations" presents some weaknesses of the present study in a very candid way.
Response 7: Dear reviewer, thank you for your positive comments!
Comments 8: CONCLUSIONS chapter: the aim of this study is achieved and the results are well presented.
Response 8: Dear reviewer, thank you for your positive comments! And we kindly appreciate all your efforts and support on our current manuscript!
Reviewer 2 Report
Comments and Suggestions for Authors
The work presented to me for review is interesting, but there are some ambiguities in it:
1. The authors mark physical activity as appropriate and inappropriate - what does this mean, on what basis was it determined, was a test used to assess physical activity?
2. Was the sample size determined, how was the cohort size determined, is it consistent with the population structure?
3. Who completed the surveys, were reading comprehension skills taken into account in the group of the youngest respondents, were the surveys completed simultaneously with anthropometric measurements?
4. Why was the study divided into study durations above and below 2 hours - the studies presented in this article do not require such a long time?
5. Why was only the mother's education taken into account in the survey?
6. Was the same equipment used to measure blood pressure in children and adolescents - blood pressure monitors with special measuring cuffs adapted to the reduced circumference of the child's arm are dedicated for children?
7. The methodology does not indicate how blood pressure is determined and abdominal and general obesity were differentiated
8. in the methodology the authors use the terms eastern, western ... and in the discussion they compare two provinces defining them by name - this should be unified
9. in the article the authors draw attention to the differentiation of blood pressure depending on sexual maturity, however they do not provide information on how it was defined
10. in the introduction there is no information regarding previous studies on overweight and obesity or hypertension in children and adolescents in China
Author Response
Dear reviewer, thank you for spending your precious time on our manuscript. We kindly appreciate all your efforts! Regarding your concerns related to this manuscript, we have revised them according to the following comments:
Comments 1: The authors mark physical activity as appropriate and inappropriate - what does this mean, on what basis was it determined, was a test used to assess physical activity?
Response 1: Dear reviewer, thank you for your question. Our current study surveyed the duration of daily moderate-to-vigorous physical activity (MVPA, activities that could cause shortness of breath or sweating, including jogging, biking, swimming, doing housework, etc.) among the respondents. We summed up the daily average duration of all the above MVPAs to further define physical activity level. The physical activity level in the current study was defined based on the Physical Activity Guidelines for Chinese (2021), which standard was "having at least 60 minutes of moderate to vigorous physical activity per day". Hence, if the daily average duration of MVPAs is over 60 minutes, we define it as "adequate" and the contrary as "inadequate. "
To further clarify the definition, we also updated the description in "2.5 Statistical analysis" of the manuscript in Lines 169–171, "daily average duration of moderate-to-vigorous physical activity achieved over 60 minutes, adequate/inadequate".
Comments 2: Was the sample size determined, how was the cohort size determined, is it consistent with the population structure?
Response 2: Dear reviewer, thank you for your question. Based on your concerns about sample size and national representativeness, we have revised the corresponding statements in the "2.1 Participants" section in Lines 78–80, "The sample size of the respondents aged 6~17 years was based on the prevalence of overweight (4.5%) in 2013 and the non-response rate (10%). Finally, 71,035 respondents were sampled." And in Lines 80–81, "The first stage of sampling was to select 275 survey sites randomly based on the considerations of regions and urbanization."
Furthermore, we also revised the statement in "2.5 Statistical Analysis" in Lines 196–199: "The survey weight used in this study was calculated based on the data published by the China National Bureau of Statistics in 2010, including post-stratification weights and sampling weights, to obtain national representativeness and the same population structures between survey samples and the national population."
Comments 3: Who completed the surveys, were reading comprehension skills taken into account in the group of the youngest respondents, were the surveys completed simultaneously with anthropometric measurements?
Response 3: Dear reviewer, thank you for your question. Based on your concerns, we have revised the statements in "2.2 Information collection" in Lines 124–126: "Given the consideration of reading comprehension skills, the information of respondents in primary schools was collected by interviewing their parents/caregivers, whereas the information of respondents in junior high schools was collected by interviewing them-selves." And in "2.3 Anthropometric and blood pressure measurement" in Lines 134–135: "All participants underwent anthropometric measurements according to standardized protocols by well-trained staff in the morning on an empty stomach during the investigation period."
Comments 4: Why was the study divided into study durations above and below 2 hours - the studies presented in this article do not require such a long time?
Response 4: Dear reviewer, thank you for your question. In our current manuscript, we did not mention study durations. I am afraid that you mean the "screening time"? For screening time, we found it may cause misunderstanding and have already revised it as "video time." It has been reported that high video time was associated with systolic blood pressure, so we adjusted video time in the current study. Based on the previous study, it was defined as the total time spent on TV, smartphone, or other types of electronic screens within a day and categorized as < 2 h/≥ 2 h based on the previous study [1]. To further clarify, we revised "screening time" to "video time" in the whole manuscript and also revised the statement in "2.4 Outcomes definitions" in Lines 171–172: "video time (daily average duration of spending on TV, smartphone, or other types of electronic screens within a day, < 2 h/≥ 2 h)" and added the reference.
Comments 5: Why was only the mother's education taken into account in the survey?
Response 5: Dear reviewer, thank you for your question. As reported in the previous study, the lower the education of the mother, the higher the SBP and DBP of their children [2]. Thus, in the current study, we also included maternal educational level as a confounding variable and categorized it as primary school or below/junior high school/senior high school or above. To further clarify the statement, we added the reference in Line 168.
Comments 6: Was the same equipment used to measure blood pressure in children and adolescents - blood pressure monitors with special measuring cuffs adapted to the reduced circumference of the child's arm are dedicated for children?
Response 6: Dear reviewer, thank you for your question. In this version, we have revised the statement in "2.3 Anthropometric and blood pressure measurement" in Lines 145–148: "Moreover, each sphygmomanometer was equipped with three types of measuring cuffs, namely SS size (12~18 cm), S size (17~22 cm), and M (22~32 cm). Investigators were asked to select the appropriate size of measuring cuff based on the respondent's age and upper arm circumference."
Comments 7: The methodology does not indicate how blood pressure is determined and abdominal and general obesity were differentiated
Response 7: Dear reviewer, thank you for your question. For blood pressure, we used the 3-time average value of systolic blood pressure and diastolic blood pressure to represent respondents' blood pressure status. In the manuscript, we have stated in "2.3 Anthropometric and blood pressure measurement" in Lines 150–152: "Eventually, the systolic blood pressure (SBP) and diastolic blood pressure (DBP) were rep-resented by three-time average measurements of SBP and DBP." Meanwhile, for abdominal obesity and general obesity, we revised the definition this time. The former was stated in "2.4 Outcomes definitions" in Lines 162–164: "For general obesity, the participants were classed as normal, overweight, and obesity based on age- and sex-specific BMI percentile tables from the recommendations of the Working Group on Obesity in China." And the latter was in Lines 164–165: "Central obesity (abdominal obesity) was defined according to whose WHtR>0.46."
Comments 8: in the methodology the authors use the terms eastern, western ... and in the discussion they compare two provinces defining them by name - this should be unified
Response 8: Dear reviewer, thank you for your suggestion. In Table 3, we presented the weighted prevalence of HTN and pre-HTN by geographical regions or provinces. Hence, to be unified between methodology, results, and discussion, we have revised the corresponding sentence in Lines 406–410: "Central region showed to have the highest prevalence of HTN and pre-HTN compared with the prevalence in Eastern and Western regions. However, when compared by provinces, Hebei province, which was in the Eastern region, had the highest HTN prevalence, whereas Anhui province, which was in the Central region, had the highest pre-HTN prevalence."
Comments 9: in the article the authors draw attention to the differentiation of blood pressure depending on sexual maturity, however they do not provide information on how it was defined
Response 9: Dear reviewer, thank you for your suggestion. The current study defined HTN and pre-HTN per year of age, sex (males or females), and different height groups, given different statuses of sexual maturity based on the 2017 updated blood pressure references for Chinese children and adolescents aged 3~17 years. Moreover, to further clarify the statement, we have now added the statement in "2.4 Outcomes definitions" in Lines 156–158: "Specifically, they were defined by per year of age, sex (males or females), and different height groups." The reference was cited in the corresponding sentence [3].
Comments 10: in the introduction there is no information regarding previous studies on overweight and obesity or hypertension in children and adolescents in China
Response 10: Dear reviewer, thank you for your suggestion.
In this version, we have added more information regarding elevated blood pressure and hypertension among Chinese children and adolescents and updated the introduction section in Lines 47–63: "HTN in adulthood can originate in childhood. Studies have shown that individuals with persistently elevated blood pressure from childhood to adolescence are more likely to develop self-reported HTN in adulthood compared to those with normal blood pressure. In 2010, the prevalence of elevated blood pressure among children and adolescents aged 7 to 17 in China was 16.1% for boys and 12.9% for girls. Over the past two decades, the prevalence of HTN in children and adolescents has been increasing, and currently, millions of children worldwide have elevated blood pressure. Another study also reported that around 4% of Chinese urban children and adolescents aged 6~17 years were confirmed to have hypertension. However, the onset of HTN in childhood and adolescence is difficult to detect because the symptoms of elevated blood pressure have not yet manifested during this period. Previous studies have shown that HTN and pre-hypertension (pre-HTN) were frequently undiagnosed in the pediatric population. Therefore, understanding the prevalence of HTN-related status among Chinese children and adolescents is essential for developing new strategies for future HTN prevention, especially at a nationwide level. Furthermore, the factors influencing HTN among Chinese children and adolescents are not fully understood, which is key to HTN identification and management. Thus, identifying related influencing factors is equally urgent currently."
Moreover, as overweight and obesity served as potential influencing factors, we focused this part on the discussion section in paragraph 5 and have revised the statement in Lines 451–454: "As overweight and obesity are spiking among Chinese children and adolescents aged 6~17 years, the prevalence has reached 11.1% and 7.9% for them, respectively. This condition has posed a significant threat to HTN-related status among Chinese children and adolescents."
Reference
1. Giussani, M.; Antolini, L.; Brambilla, P.; Pagani, M.; Zuccotti, G.; Valsecchi, M.G.; Lucini, D.; Genovesi, S. Cardiovascular risk assessment in children: role of physical activity, family history and parental smoking on BMI and blood pressure. J Hypertens 2013, 31, 983-992, doi:10.1097/HJH.0b013e32835f17c7.
2. Bouthoorn, S.H.; Van Lenthe, F.J.; De Jonge, L.L.; Hofman, A.; Van Osch-Gevers, L.; Jaddoe, V.W.; Raat, H. Maternal educational level and blood pressure, aortic stiffness, cardiovascular structure and functioning in childhood: the generation R study. Am J Hypertens 2014, 27, 89-98, doi:10.1093/ajh/hpt180.
3. Fan, H.; Yan, Y.; Mi, J. Updating blood pressure references for Chinese children aged 3-17 years. Chin J Hypertens 2017, 25, 428-435, doi:10.16439/j.cnki.1673-7245.2017.05.009.